# Implementing a Physical Activity Promoting Program in a Flex-Office: A Process Evaluation with a Mixed Methods Design

**DOI:** 10.3390/ijerph17010023

**Published:** 2019-12-18

**Authors:** Viktoria Wahlström, Anncristine Fjellman-Wiklund, Mette Harder, Lisbeth Slunga Järvholm, Therese Eskilsson

**Affiliations:** 1Department of Public Health and Clinical Medicine, Section for Sustainable Health, Umeå University, 90187 Umeå, Sweden; lisbeth.jarvholm@regionvasterbotten.se; 2Department of Community Medicine and Rehabilitation, Section of Physiotherapy, Umeå University, 90187 Umeå, Sweden; anncristine.fjellman-wiklund@umu.se (A.F.-W.); therese.eskilsson@umu.se (T.E.); 3Umeå School of Architecture, Umeå University, 90187 Umeå, Sweden; mette.harder@umu.se

**Keywords:** activity-based work, ergonomics, longitudinal, office design, sedentary behavior, office workers, workstation

## Abstract

The aim of this study was to investigate facilitating and hindering factors when implementing a physical activity (PA)-promoting program among office workers moving to a flex office, by conducting a process evaluation. Additionally, we evaluated self-reported and perceived PA behaviors. With a mixed methods design, analyses were based upon data from interviews with 70 employees and repeated questionnaires from 152 employees. The PA-promoting program was fully implemented and supported by management. There was a strong health promoting culture, encouraging PA in the organization already at the start of the study. The lecture and the office design were rated as the most motivating program components. The use of stairs, breaks during meetings and social acceptance for standing and walking at work increased. Employees described a strive for variation, and how managers, the office environment, productivity and ergonomic aspects influenced sedentary behavior (SB) and PA. The need for the PA-promoting program was questioned, and the timing of the program was debated. To conclude, a strong organizational health culture combined with a facilitating physical environment can create sustainable positive PA behaviors in office settings. A thorough understanding of organizational needs and a participatory process are needed to tailor organizational interventions to decrease SB.

## 1. Introduction

Due to advancing technical developments, working life has changed enormously during the last decades, and over the last 50 years, the proportion of workers with sedentary work tasks has greatly increased [1]. Sedentary behavior (SB), defined as any waking behavior in a sitting, reclining or lying posture with an energy expenditure ≤ 1.5 metabolic equivalents [2], has been reported to be a risk factor for type 2 diabetes, cardiovascular disease and early mortality [3,4]. Studies have shown that employees in offices are seated for up to 80% of their work time; thus it is of public health importance to decrease SB and increase physical activity (PA) among office workers [5,6,7,8].

Multicomponent interventions among office workers have shown the greatest reductions in workplace sitting, followed by environmental and educational strategies [9]. Recent cluster randomized trials in office settings [7,10,11], including sit-stand workstations in combination with multilevel strategies, have shown reductions in sedentary time, that are likely to lead to improvements in outcomes like weight and waist circumference, but the knowledge of the long-term efficacy and the cost-effectiveness of these interventions is still limited [12]. As there are variations in workplace context, i.e., in terms of workplace sector, organizational culture and structure, an intervention that has shown to be effective in one context, might not work likewise in another setting. Therefore, there is a need of studies that facilitate the knowledge of how to transfer intervention results to different office settings [13].

It is suggested that innovative office designs, such as flex offices, could lead to decreased SB and increased PA, but evaluations of the effects of such changes are sparse. In recent years, flex offices with activity-based working (ABW) have increased in popularity [14,15]. The main drivers for this office type are decreased facility costs and suggested facilitation of communication and teamwork within the organization [16,17,18]. A flex office provides different types of environments, for example open landscape areas, touch down workstations, cell office rooms, conference rooms of different sizes and break out spaces. Employees do not have personal workstations, but rather, choose an area which suits the current work task [15], and as employees are supposed to move between workstations, a transition to a flex office with ABW might theoretically increase variation and PA. There is, however, limited evidence that flex offices increase PA, and more studies with objective measurements are needed [16]. Hallman et al. [19] used accelerometer-based assessments to evaluate SB in an organization that relocated employees at four office sites to flex offices. The study showed increased walking time after relocation, but no changes in sitting or standing time was seen. The results differed between office sites, suggesting that contextual factors like organizational culture and office design could be of importance for the outcomes.

When planning and evaluating interventions aiming to decrease SB in office settings, Mackenzie et al. [13] highlight the need of a participatory approach in intervention development and a comprehensive process evaluation (PE) of the intervention. PE studies, preferably with a mixed methods approach, can provide a deeper understanding of facilitating and hindering factors [20,21]. Detailed contextual descriptions facilitate the interpretation of results which enables the understanding of the link between the intervention activities and intervention outcomes [13,20]. As flex offices are becoming more common, there is a need to further explore and evaluate the effects of interventions aiming to decrease SB and increase PA in this office type. A few studies [7,10,22,23] have presented quantitative results, reporting intervention dose delivered and received, but to our knowledge, no previous studies have used mixed methods to conduct a PE of an intervention aiming to decrease SB and increase PA among office workers.

We have previously published results from the Active Office Design (AOD) study, with objectively measured SB and PA in two office groups; one of the groups relocating to a flex office with ABW, and the other to an office with a traditional cell office design [24]. In both groups, a multicomponent PA-promoting program was implemented in relation to the relocation. In short, we found no changes in sitting or standing time at work after relocation, but an increase in walking time, and moderate to vigorous PA at work in the flex office group compared to the cell office group.

The first aim of this study was to investigate the facilitating and hindering factors when implementing a PA-promoting program in a group of office workers relocating to a flex office, by conducting a PE. The second aim was to evaluate self-reported and perceived PA behaviors before and after the PA-promoting program.

## 2. Materials and Methods

This study has a mixed methods design [25], which means that quantitative and qualitative data were collected and analyzed in parallel, but separately (Figure 1). Data were integrated during the interpretation and reporting of results, by integrating through narrative using a weaving approach [25].

### 2.1. The Overarching Research Project—The Active Office Design Study

The current study is part of the Active Office Design (AOD) study; a longitudinal intervention study with the overall aim to evaluate the effects on work environment, productivity, physical activity (PA), health and wellbeing when relocating office workers to a flex office. In the AOD study, a municipality administration was followed during the relocation to new premises, from 6 months before to 18 months after relocation. A group of 228 office workers from the economy, human resources, urban planning and education departments, and a group of politicians were relocated from cell offices to a flex office with activity-based working (ABW). The office workers carried out a variety of work tasks, and had a high level of autonomy with respect to planning and performing the work. The studied organization was servicing a municipality with 56,000 inhabitants, and carrying out broad assignments, like running and developing schools, social services and urban planning. As a whole, the organization employed around 5000 persons, was politically governed and publicly financed.

### 2.2. Development and Implementation of the Physical Activity Promoting Program

In parallel with the relocation, a PA-promoting program initiated by the researchers was developed and implemented. In order to fit the intervention program to the context, we applied a participatory approach with collaboration between researchers and workplace representatives. The PA-promoting program aimed to decrease sitting, increase standing and walking, and break up prolonged sitting.

Before relocation, all employees had individual workstations, adjustable chairs and sit-stand tables. In the new flex office, employees had shared workstations instead. All workstations had adjustable chairs, computer screens and sit-stand tables. The new office building had three floors with predominantly open office landscape areas as well as additional cell office rooms and different sized conference rooms. Each floor had seven centrally placed waste-paper bins and two printer rooms. An open staircase connecting all floors was centrally placed, right inside the main entrance. When planning the relocation, organizational representatives, the researchers and an ergonomist from the occupational health service, discussed how the interior design could be further developed to facilitate PA in the office. This led to furnishing with both sitting and standing tables in the break out spaces, sit-stand tables in some of the meeting rooms, and most of the touch down stations mainly consisting of standing tables. The office was also equipped with 16 treadmill workstations, placed in both shared cell offices and open landscape areas. One month after relocation, about 5% of the employees participated in a drop-in session, where one of the researchers (VW) introduced the treadmill workstations. A folder with information on how to use the treadmill, including ergonomic instructions, were placed at the shared treadmill workstations.

In addition to the adaptation of the physical environment, the PA-promoting program consisted of a lecture about sedentary behavior (SB) and PA for all employees, a workshop with managers and a three-step communication campaign. The PA-promoting program thus included components for environmental, organizational, group and individual factors, and activities were implemented over a period of 18 months (Figure 2). The communication campaigns were developed at four workshops that included brainstorming and discussion among ‘health inspirers’ and a health strategist, all from within the organization. The role of the health inspirers was, together with the managers, to support and inspire their coworkers to a healthy lifestyle. 

The communication campaigns focused on (1) interruption of prolonged sitting, (2) importance of everyday PA like taking the stairs, active commuting or walking meetings, and (3) the usage of treadmill workstations in the office. The organization’s communication department and the internal health strategist delivered all material via posters, table-tops in meeting rooms and break out spaces, as well as posts on the workplace intranet. Information to managers were communicated via manager meetings and e-mails, and they were asked to disseminate and discuss the messages from the campaigns at workplace meetings. More details on the theoretical background and content of the PA-promoting program have been described elsewhere [24]. About three months after each data collection period, the main results from the focus groups and the questionnaires were presented at the workplace via oral presentations, where all employees were invited.

The AOD study received ethical approval from the Regional Ethical Committee in Umeå, SE (No: 2014/226-31). Separate information and consent forms were provided for the questionnaire and interviews of the study; all participants signed an informed consent form.

### 2.3. Data Collection

#### 2.3.1. Qualitative Data

Qualitative data were collected through focus groups and individual interviews (Figure 2). During the study, organizational information was collected at regular quarterly meetings between researchers and the organization’s relocation steering group which gave an understanding of things happening within the organization not related to the PA-promoting program, that might affect outcomes [26]. Data were collected from October 2014 to October 2017.

All employees involved in the relocation were invited via e-mail to participate in focus groups conducted approximately at 6 and 18 months after relocation [27]. A convenient sampling from these invitations was applied. The interviews lasted for 50–90 min, involved 3–5 participants, and were led by two researchers who were not otherwise involved in the implementation of the PA-promoting program or the current study. The interviews were conducted using a semi-structured interview guide, and covered overall experiences of the relocation process. Specific questions related to the current study were included, and these focused on perceived barriers and facilitators for SB and PA and experiences of the performed PA-promoting program. Examples of questions were how the physical environment, co-workers and managers influenced PA behaviors. One question addressed perceptions of the different components of the PA-promoting program. After performing all focus groups, a need for complementary data was identified to meet the criteria in the process evaluation (PE) model and framework [19,21]. Therefore, additional invitations were sent via e-mail to managers and key persons to find out more about their roles in relation to the work environment and the PA-promoting program. At 20 months after relocation, individual interviews with six managers were performed by the same researchers performing the focus groups. These interviews took 30–45 min, and included questions about experiences and management related to the PA-promoting program. One interview with two key stakeholders focused specifically on the perception of the development and implementation of the PA-promoting program, the relation to the researchers, as well as their learning experiences from participating in the study. This interview was performed by one of the authors (TE). All interviews were recorded and transcribed verbatim, some by one of the researchers in the group, some by the first author (VW), and some by University students. VW listened to all interviews to ensure that the transcriptions were correct.

#### 2.3.2. Quantitative Data

All employees were invited to fill in questionnaires about perceived working conditions, productivity, health and PA at 6 months before (baseline) and 18 months after the relocation. Coded questionnaires were distributed via managers together with a sealable return envelope. Employees could fill in the questionnaires during working hours.

Background characteristics: At baseline age, gender, managerial position and employment degree were reported. Time spent with computer work was assessed on a 4-graded scale from ‘0–2 h’ to ‘6–8 h’. Stress and musculoskeletal neck–shoulder symptoms were assessed for the last 3 months on a 5-graded scale from ‘Never’ to ‘Always’ [28]. Self-rated general health was assessed by a question from the 36-Item Short Form Health Survey which has shown strong reliability and validity [29]. Exercise habits were reported by using the question: “*How many days during the last 3 months have you exercised in workout clothes, with the purpose of improving your fitness and/or to feel good?*” This question has been previously used in studies [30,31].

Measurements: To measure self-rated PA behaviors and perceptions of the different possibilities to be active at work, we adapted questions from a previous study [32], since no validated questions were available. The usage of different possibilities to be active at work was assessed by six items (standing while working, standing at meetings, taking the stairs, participating in walking meetings, walking or cycling to meetings outside the office and using treadmill workstations), each with a 5-point scale from ‘never’ to ‘daily’. Three items assessed the habits of taking breaks during meetings, suggestions for walking meetings and walks during breaks, on a 4-point scale from ‘never’ to ‘often’. The perceptions of using the different possibilities to be physically active at work were assessed by six items on a 4-point scale from ‘do not like at all’ to ‘like a lot’. Social acceptance and whether clothes were a barrier to standing or walking at work were assessed on a 4-point scale from ‘not at all’ to ‘to great extent’. Questions about the motivational impact of the different activities in the PA-promoting program were included in the questionnaire at 18 months, using five questions, (e.g., To what extent did the communication campaigns motivate you to sit less and move more at work?) with assessments on a 4-point scale from ‘not at all’ to ‘to great extent’.

#### 2.3.3. Description of the Process Evaluation Model

For the process evaluation we used a modified model, based on a PE model and framework developed by Nielsen et al. [21,26]. (Figure 1). The model is influenced by disciplines such as participatory ergonomics and organizational development [21,26], and includes elements for context, the intervention and mental models. For each element, an associated framework includes questions of importance for data collection and the presentation of results. 

The element on context is divided in ‘omnibus context’, which includes overall organizational information, description of culture and norms of importance for the intervention, and the ‘discrete context’, aiming to describe occurring events, within or outside the organization during the intervention, which might have hindered or facilitated the effects [21]. The ‘intervention element’ highlights initiation, development, intervention planning, implementation strategies and the involvement of leadership, managers and consultants. ‘Mental models’ aim to capture whether different ‘actors’ in the organization were ready for change, and in what way the intervention activities were perceived and interpreted [21]. To meet the second aim, to evaluate self-reported and perceived PA behaviors, we modified the model by adding a fourth element, ‘behaviors’. This element focused on how employees were physically active at work, for example how they described patterns for sitting, posture variation or different possibilities to walk at work. The fourth element also included descriptions of barriers and facilitators for PA, such as musculoskeletal symptoms, the possibility to tune the workstation ergonomics for individual needs, as well as descriptions of perceived effects related to sitting or moving at work.

#### 2.3.4. Analysis of Qualitative Data

Qualitative data were analyzed using a deductive Qualitative Content Analysis (QCA) approach [33] with the modified model and associated framework as a structured matrix [21] (Figure 1). The analysis started with a first reading of the transcribed interviews to get an understanding of the content of the data material, after which the coding was performed. The data analysis was performed in the following way. During all steps of the analysis three of the authors (VW, TE, AFW) coded the transcripts independently, followed by a mutual comparison and a final negotiated outcome between all of the authors [34]. The first author (VW) extracted parts of the transcripts of relevance to the study aim. Meaning units from the transcripts were condensed, and abstracted into codes. The codes were then sorted deductively into the four elements in the PE model. Following that, the first author brought the codes together on a more abstract level into preliminary subcategories and categories, guided by the PE model and its associated framework. Finally, these preliminary subcategories and categories were discussed, reflected on and condensed into final categories. A category refers mainly to a descriptive level of content and answers the question: “*What?*”, and can thus be seen as an expression of the manifest content of the text. To strengthen the trustworthiness of our analysis and verify the interpretations, preliminary results were presented to all AOD study researchers who were well versed in the study, and their feedback guided further discussion and analysis [34,35].

#### 2.3.5. Analysis of Quantitative Data

Wilcoxon matched paired tests were used to test for changes between baseline and 18 months. Descriptive statistics were used for perceived motivation from the PA-promoting program at 18 months. Statistical analyses were performed using SPSS software v.24 (IBM Corp, Armonk, NY, USA), and the significance level was set at α = 0.05.

## 3. Results

### 3.1. Participants and Background Characteristics

At baseline, 228 employees who moved to the flex office received a questionnaire, and 219 (96%) responded. At 18 months, 152 out of 171 employees (89%) responded. In this study we analyzed questionnaire data from the 152 employees, responding at both baseline and 18 months after relocation (Table 1). Reasons for drop-out between the baseline and the follow-up were employee turnover (*n* = 40), retirements (*n* = 6), changes of office relocation (*n* = 3), parental leave (*n* = 4), or sick leave (*n* = 4). Of the respondents, 102 (67%) were women and 50 were men, 26% managers, 53% over 50 years of age and 60% rated their health as very good or excellent (Table 1). At baseline there was a difference between men and women for time spent with computer work per day (*p* = 0.018), and habits for physical exercise (*p* = 0.020). No differences for background variables at baseline were seen for age, when comparing employees between 18–49 and over 50 years. 

In total, 70 individuals, (53 women and 17 men) 30–63 years old, participated in interviews, whereof 16 were managers. In total, 12 focus groups were carried out. At six months after relocation a separate focus group with managers was performed. Fifteen employees participated in focus groups at both 6 and 18 months after the relocation. Due to challenges to organize a focus group for managers, six individual interviews with managers were performed 20 months after the relocation. Four of the managers participated in both focus group and individual interviews. Two key persons, a health strategist who was the contact person for the PA-promoting program in the organization during the study, and a senior manager responsible for the office relocation, participated in the additional interview at 24 months (Figure 2).

The results are presented in relation to the modified PE model (Figure 1) [21]. The presentation of results combines qualitative results, with interwoven citations and questionnaire data [25]. Subcategories and categories in relation to the elements in the PE-model are given in Table 2.

### 3.2. Context of the PA-Promoting Program

#### 3.2.1. Support for Physical Activity

Managers and employees described a long-standing tradition within the organization to systematically work with occupational health and encourage health promotion, including awareness of posture variation and exercise. For example, the organization had a systematic health-promoting program, including a policy (common in Sweden), that all employees could exercise one hour a week during working hours. The studied organization provided subsidized gym fees for their employees to use external gyms. Bicycles and electric bicycles were provided for active transport to meetings outside the office. To facilitate PA, showers were available at the workplace. Under the course of the study, the organization also implemented easy access to a computer program using push notifications as a reminder to take active breaks. The timing of the notifications and exercises could be adapted to individual preferences. The organization also provided Pilates-balls and ergonomic stools without backrests in the office.

#### 3.2.2. Environmental and Ergonomic Challenges

During the study period, many refugees arrived in Sweden, and to manage the reception of refugees, the organization hired more staff. The consequence was that the office became crowded, and sometimes it became difficult to find a suitable workstation. Overall, employees thought that the flex office was equipped to create good ergonomic conditions, but the ergonomic adjustments of chairs were perceived as time consuming and difficult, described as “*one cannot spend ten minutes adjusting the chair, and become frustrated before starting to work*”.

### 3.3. Intervention Development and Implementation of the PA-Promoting Program

#### Clarity in the Organization

All activities in the PA-promoting program were delivered as intended. The key stakeholders perceived the intervention program as well planned, with clear definitions of roles and an open dialog between researchers and representatives from the organization. In the workshops, where communication campaigns were developed, 2–7 ‘health inspirers’ participated, which were fewer than expected. Employees expressed that there was a lot of communication around PA, which suggests that information had reached out. Regarding responsibility, employees described how “*we have a shared responsibility to be active but managers go first*”. Most managers had acted like role models and supported the program by encouraging the usage of sit-stand tables, the utilization of the health and wellness hour, and by discussing health and PA at unit meetings. In the questionnaires, 35% of employees reported that their managers’ behavior had been a motivating factor to sit less and move more to some extent (Figure 3). Managers expressed that the expectations for them to encourage the PA-promoting program were clear from top management. They thought their roles to influence norms and attitudes were important, but reflected on how far one should go to get the message through, because “*it is still up to every individual to make a change, it is difficult, I can only remind them of their possibilities, just like the campaigns*”. Concerning motivation and responsibility, both employees and managers expressed how the organization can provide possibilities to increase PA, but in the end, it is the individuals’ choice to use them.

The existing health-promoting program in the organization was perceived to mainly emphasize PA, and the PA-promoting program was described to further reinforce this focus. At baseline 71% rated standing or walking to be socially accepted to a great extent, and at 18 months this rating had increased to 84% (*p* < 0.001). Combined, these results reflected an increased social acceptance in the organization (Table 3). The quote “*I wonder if health promotion automatically must be physical activity? For me it’s not!*” represented a wish for a more balanced health promoting program, to meet all the needs in the organization. Stress coping activities like yoga- or meditation were suggested as warranted activities. The employees reported the motivational impact of the PA-promoting program components in the 18 months questionnaire. The perceptions varied, but the office design and the lecture had highest scores, while communication campaigns had the lowest (Figure 3). Sixty-eight percent participated in the lecture, and among those 32% thought it was quite motivating or motivating to a great extent, while 44% thought it was motivating to some extent. Varying opinions were also reflected in the interviews, where some did not think they learned anything new, while others appreciated a deeper understanding of the physiological reasons for, i.e., breaking up prolonged sitting, and the importance of PA in everyday life.

### 3.4. Behaviors

#### Voluntary and Involuntary Physical Activity and Mediators

Employees thought that the office design provided great opportunities for standing, and many described how they strived to alter between sitting and standing, appreciating the variation. Others preferred sitting at work, and described it as comfortable and a habit. Some mentioned that they exercised enough during leisure time, and they did not think standing at work was needed for them. Barriers for standing could be musculoskeletal discomfort, i.e., pain in the lower back or feet, but also an awkward feeling of ‘standing out’ when working in the open landscape areas. Questionnaires showed no significant changes for standing from baseline to 18 months follow-up, with 52% reporting that they worked standing more than three times per week at baseline and 47% at 18 months. At the same time the ratings for the appreciation of working while standing decreased significantly from baseline to 18 months (*p* = 0.003) (Table 4).

At six months follow-up employees described how prolonged sitting regularly occurred at meetings, and that breaks from sitting were uncommon, but at 18 months, breaks were described as more frequent. Confirming results were found in the questionnaires, where breaks at meetings were reported to occur more often at 18 months compared to the baseline (*p* = 0.002) (Table 3). A barrier for breaks during meetings was the difficulty to remember to take breaks when being focused and engaged. Some employees described that they had installed and used the program for active breaks when working individually by the computer. Barriers for usage were inconvenience in relation to work tasks, but also a feeling of awkwardness from suddenly doing exercises among others. Employees described being well aware of colleagues’ behaviors and how they influenced each other. The culture and norms seemed to differ between different units or groups, which affected the behaviors, expressed as “*our manager has explicitly encouraged us to use the program, and if everybody does it, it becomes normal*”. This was also reflected in the questionnaire at 18 months, where 48% reported that other workers’ behaviors had been motivating ‘to some extent’, and 22% thought that other workers’ behaviors had been ‘quite motivating or motivating to a great extent’ (Figure 3).

Experiences from the treadmill workstations varied greatly, and many described they had tried them once or more, but only a few kept using them regularly. Questionnaires showed that 12% of the employees used the treadmills at 18 months after relocation (Table 4). By the end of the study eight treadmills were still in use. Barriers for usage were noise, which led to avoidance, as not to disturb colleagues. Some perceived motion sickness after usage, “*I have tried. It worked but I got dizzy when I stopped. And I stood still for a while and the whole world kept walking on a treadmill. I knew I stood still, but it did not feel like it*”. Difficulties to find suitable work tasks for the treadmill were mentioned. Appropriate work tasks for the treadmill were proposed, and reading, checking e-mails or walking while having a phone meeting were mentioned as suitable. 

Treadmills in single office rooms were seldom used, because the priority when using a single room was to perform concentrated work, which was not found compatible with walking. Due to the experience of productivity loss, some did not prioritize to use the treadmill workstation, as it took a few minutes to get installed, described as “*you don´t choose the treadmill workstation as you want to prioritize work. Maybe you choose to stand by the desk instead*”. The overall lack of workstations was also mentioned as a barrier to use the treadmills, because it could be difficult to find a workstation after walking. A reflection among employees was that the treadmills were a good idea, but that there were too many treadmills to begin with. A few employees were irritated and thought that treadmills in the office was a ridiculous idea that people outside the organization rattled about.

As described above, the office became crowded, and employees described spending time walking around the office to search for a workstation or for locating colleagues. This was perceived as both time consuming and frustrating, “*it bothers me to walk around on my work time. I want to get started in the morning, and not waste time to find a workstation*”. To avoid this, some employees arrived to the office early in the morning to get a suitable workstation, which they would usually use the whole workday. For some, the resistance to adjust the chairs became a motivation and a driver for standing while working. Pilates-balls and ergonomic stools were used as an alternative to standing, and were described as comfortable and easier to use because they did not require any ergonomic adjustments.

Some employees expressed that “*there is no culture for walking meetings in our organization*”, although many had participated in at least one, starting with a walk and talk and ending sitting down to take notes. Some also described “*we tried in the beginning, just a few times. But we stopped. It was great fun when we did it, but then we just didn´t continue*” and they thought “*walking meetings work best when being two persons. It is actually the easiest*”. Most of these walking meetings were performed at the annual performance appraisals with their manager. Questionnaires showed an increase of suggestions for walking meetings between baseline and the 18 months follow-up (Table 3), but it still rarely occurred (Table 4). Main barriers for walking meetings were poor weather or the need for computer access during meetings. Some expressed that walking meetings felt strange, and thought it would be difficult if sensitive issues were to be discussed, while others experienced walking meetings as more relaxed for sensitive discussions, compared to sitting opposite to each other. A wide variation of perceptions of walking meetings was also reflected in the questionnaires (Table 4).

Employees described their usage of the stairs in the new office as natural, as they were placed right in front of the main entrance and were open and inviting. In addition, the placement of the elevator was discrete, around the corner from the stairs. Questionnaires showed a significant increase in the usage of stairs, with 72% taking the stairs daily at baseline, and 93% at 18 months (*p* < 0.001). Also, taking the stairs was more appreciated in the new office (*p* < 0.001) (Table 4).

### 3.5. Mental Models

#### 3.5.1. Balance of Communication Intensity

The motivation and readiness for change varied, although most participants in the interviews were positive to striving for more PA at work. The increased awareness and knowledge about SB and PA were described as something positive, and communication campaigns were described as a natural follow up of the lecture “*rubbing the message in*”, and giving examples on how to transfer awareness to practice. Others were irritated, and thought the campaigns were intrusive and provoking, described as “*and it is so important that we exercise, but I think it is my private issue. I get annoyed*”. A few employees thought there had been too much focus on PA, and questioned the reasonableness as “*you are at work to work, not to be physically active*”.

Some described what they thought to be an unbalanced focus of performed health activities on PA, which did not match all of the actual needs in the organization. They were of the opinion that there had been too much focus on PA, and would have preferred to focus more on developing the use of the office, such as availability of workstations, rules and zoning. 

The challenges to perform ergonomic adjustments when changing workstation was perceived as a deterioration of the ergonomic work environment, and a few employees interpreted the PA-promoting program as a way to compensate for this deterioration, expressed as “*is this to compensate for the bad work environment?*”

Further, there were reflections on the difficulty to find an appropriate level on the amount of information, “…*and the interest is high in the beginning, then you easily go back to old behaviors. So, it is important to repeat. But if it becomes too much, people that are not interested can turn their backs to it…so it is important to find the right level*”. In relation to communication intensity, aspects of discrimination were also highlighted, “*physical activity is all good, but it cannot rule out the message that everybody should feel welcome in this organization*”, and a balanced communication was discussed as both important and challenging.

#### 3.5.2. Openness for Activity

Overall, employees described, more or less explicitly, a strive for posture variation and to use the possibilities to be more active at work, but without compromising productivity. Some also thought that integrated PA at work might lead to increased alertness, and thereby have positive influence on productivity. By reflecting on the changes of norms and attitudes, many employees thought they were slowly moving towards new ways of working. As an example, walking meetings with managers were described not to be conceivable earlier, but now many had tried it. During the focus groups, employees had open-minded discussions, and brainstormed further ideas to increase PA at work. They suggested redesigned conference rooms, office bikes and balance boards, a separate room for active breaks—equipped with treadmills, bicycles and equipment for light exercise breaks, and marked outdoor paths for walking meetings or lunch walks. The importance of continuing to develop, to try new things and to encourage managers to act like role models, were also mentioned by top management. Employees also reflected on the normalization of new behaviors, as “*new habits also work*”. As a foundation for new behaviors, trust in employees was emphasized as a facilitator, “*as long as you deliver you can do things the way you want*”.

## 4. Discussion

The first aim of this study was to evaluate the implementation process of a PA-promoting program among office workers, in order to better understand the factors that influenced the implementation process and the results. The use of a theoretical PE model and mixed methods made it possible to interpret and explain the results. There was a strong health promoting culture and frequent use of sit-stand tables in the organization already at the start of the study. We found an increase of social acceptance for standing and walking at work, an increase of breaks during meetings and increased use of stairs. The office design and the lecture were perceived as the most motivating ingredients in the PA-promoting program. The need for the program was debated, and the rationale of the program somewhat came into conflict with needs related to the relocation to a new office and new ways of working, indicating that the timing of the PA-promoting program was not optimal. To our knowledge, no previous studies have performed such a comprehensive PE of an intervention aimed to decrease SB and increase PA among office workers.

### 4.1. Finding the Appropriate and Optimal Level of Standing

Many employees reported how they often worked standing already at baseline, and no changes were seen for self-reported standing at follow-up. This result is in accordance with objective measurements within a subgroup of the studied population presented in a previously published paper [24]. Compared to other studies [7,10], employees in the current study report more standing and less sitting throughout the whole study period. A recent Dutch study evaluated the frequency of the usage of sit-stand workstations for employees with long-term access, and report that approximately 32% were non-users and 30% were daily users [36]. Our results showed that 10% were non-users and 35% daily users at the end of the study. 

Due to the long-term availability of sit-stand desks and the existing ergonomic awareness and health culture even before relocation, the stability of outcomes in our study might represent a ceiling effect for the amount that sitting time can be reduced in office workers. There is limited knowledge about the long-term efficacy of worksite interventions to decrease SB [12], and still our study shows that a facilitating physical environment and a strong health culture can create sustainable active behaviors among office workers. The habit of sitting was described as a barrier for standing, which is also described in other studies [37,38]. Interestingly, employees integrated ergonomics when talking about SB and PA, and the strive for posture variations were often mentioned. In an Australian study, occupational health practitioners underlined the importance to find a balance between sitting and standing, not to replace one workplace health issue (too much sitting) with another (too much standing) when implementing interventions to decrease SB among office workers [39]. Studies indicate that a decrease in musculoskeletal symptoms when implementing sit-stand tables are related to the possibility to freely choose how to vary between sitting and standing [40], but there are yet no guidelines for how much sitting and standing is recommended [41].

### 4.2. Walking for Good and Bad

In our previous study with objective measurements, an increase in walking time at work was seen in the flex office [24]. Results of the current study indicate that the increase in objectively measured walking time might be a result of walking to find a suitable workstation or to look for colleagues, and this might partly be due to crowdedness in the office. Even though the total square meter area of the new office was about the same as prior relocation, all employees had the possibility to use the whole office area, compared to spending most of the time in the ‘home-corridor’, which might make employees walk longer distances within the office. More walking could be positive for health, but it might also impact the experience of stress and productivity negatively. A recent Swedish study evaluating the role of environmental perceptions and use of workspaces in a flex office showed that spending more than five minutes per day looking for a workstation was associated with lower self-rated productivity [42]. To understand the effects of and driving factors behind PA in flex offices, there is a need for further studies.

Previous studies have shown that short active breaks from sitting can give positive short-term effects on blood sugar levels, especially among individuals that are less physically active [43,44]. In a flex office, the idea is to move between workstations, which theoretically could stimulate active breaks. Our results indicate that employees usually used the same workstation throughout the day, which does not support the assumption of increased breaks. Questionnaires showed an increase of breaks during meetings, which is another way to interrupt prolonged sitting. Our previous study [24], which evaluated objectively measured PA and SB, did not show any changes in the frequencies of breaks from sitting in the flex office, which might be due the fact that many employees seldom participate in meetings, and therefore changes are not reflected in the objective measurements. Likewise, Hallman et al. [19] did not find any changes in break frequencies after relocation to flex offices.

Treadmill workstations were used by 12% of the employees at 18 months follow-up. In our study, noise was a barrier for treadmill use, which is previously also described by Cifuentes et al. [45]. Thus, the placement of treadmill workstations should be carefully planned in order to avoid noise disturbances. In similarity with our results, Tudor Locke et al. [46] mentioned the ‘interruption’ and time to relocate between workstations as barriers for use of shared treadmill workstations in an office. Several employees in our study experienced motion sickness after treadmill use, which deterred further use of the treadmill workstations. To get over the motion sickness threshold, and thereby stimulating further use of the treadmill, information about this transient phenomenon should be clearly provided when introducing office treadmills. Our study showed an increased use of stairs. This result might be related to the design and placement of the stairs in the new office, which has shown to be of importance for stair use [32], rather than a direct effect from the PA-promoting program.

### 4.3. Need, Timing and Responsibility for Being Active

As described, the organization had an ongoing health promoting program, and the focus on PA was further increased by the PA-promoting program. However, employees expressed a need for supporting activities in order to help them with the challenges related to work in an activity-based manner. This suggests that there was no strong need for the PA-promoting program, and that the timing of the intervention came into conflict with other needs, which might have affected the effects of the intervention.

All activities within the PA-promoting program were performed according to plan. Roles and responsibilities for managers were clear, and both top management and middle managers supported the intervention. When developing the communication campaigns, the interest for participation among health inspirers was weak. This might have been due to workload priorities, but might also reflect that the health inspirers did not perceive an obvious need for the PA-promoting program. This indicates that it could be difficult for an organization to properly assess their need of an intervention, because their own culture and norms are normalized. In our case, a survey prior to the study might have identified the already high level of standing and the existing health promoting culture, and this might have resulted in another focus of the intervention. Renaud et al. [36] recently developed a survey with a new set of questions that could be used to tailor future intervention studies. As our study was incorporated into a research project with a broader aim, and the timeline for the relocation was predetermined by the organization, a preparatory study was not feasible.

Managers described an ambivalence on how much they should promote PA, as the final decision for behavior is up the employees. This reflection is also reported in a Dutch study, were managers described that the organization can provide opportunities, but people should make their own choices [38]. The balance of expectations from leadership and the integrity of workers is an ethical challenge important to consider in worksite health promotion. Responsibility for lifestyle behaviors in relation to worksite health promotion could be perceived differently. While employees might interpret the responsibility of their lifestyle behaviors as autonomous, employers might be more prone to equal responsibility to duty. This difference might contribute to ambivalent relationships between stakeholders [47].

### 4.4. The Challenge of Balancing the Message and Still Moving forward

The motivation and readiness to incorporate more PA at work varied among employees, but there was a general acceptance and motivation to incorporate posture variation and increase PA at work. Social norms have previously been described as potential barriers for standing in the office [48,49], but in our study there was an encouraging culture already at baseline, and it became even more pronounced during the study. Employees had open-minded discussions, and they brainstormed new ideas during the focus groups, which also indicates how they were open for further possibilities and new ways of working with the active body in mind.

Our results, where perceptions about the PA-promoting program varied a lot between individuals, indicate the complexity to find the ‘just right’ communication intensity in an organizational intervention. In our study the communications intensity was by some employees perceived too high. This might be due to the aspects of timing and conflicting needs, but the reflections about autonomy and equality were also raised. In contrast to our study, previous studies have directed their interventions to employees that signed up for the study, and individual coaching and motivational messages have been directed to participants only [7,10,11]. In organizational practice it may be more common to address all employees with health messages. Even though all activities and materials were developed and anchored with a participatory approach, and the tone of the messages was carefully chosen, these reactions might indicate that the communication intensity was too high, and the activities might have been advantageously placed at longer intervals. Our results emphasize the importance to consider the communications intensity with respect to aspects such as autonomy and equality in the communication.

### 4.5. Methodological Considerations

This study has several strengths. A long-term follow up, a rich qualitative dataset, high response rates to questionnaires, in combination with the use of a theoretical model, gave a comprehensive and structured base for analysis. To our knowledge, this is the first study on interventions to increase PA and decrease SB in office settings using mixed methods, which is recommended in PE studies [20,21,50]. The different types of data complemented each other, and the qualitative data gave possibilities for in-depth analysis [50,51]. We present an evaluation of effects in combination with an authentic and detailed description of the intervention context, which strengthens the interpretation regarding transferability [35]. Another strength is that researchers within the study represent different disciplines with expertise in both qualitative and quantitative research methods. Repeated author discussions, as well as presentation and reflections with other researchers, also strengthen the trustworthiness of the study [34].

The study also had some weaknesses. We used questions for PA outcomes that are not tested for validity and reliability, which is a possible limitation. The decision to use the PE model emerged during the study. Guidance from the framework in the planning phase of the study might have led us to include further assessments in the questionnaires. It had been desirable to have measures of possible mediators for the intervention outcomes, such as line managers’ attitudes and actions, the perceived need for the intervention, motivation to reduce SB and increase PA at work, and perceived self-efficacy for change [50,52,53]. The utilization of the health and wellness hour would also preferably have been reported from baseline. By adding items in the questionnaires and performing complementary interviews, we tried to compensate for this and complete the data. In the interviews, men, women and managers were represented, which could be assumed as a representative sample. The representation of participants might be biased, as all employees were invited to the focus groups, and employees with strong opinions might be more prone to participate. In our study, as well as in most intervention studies focusing on the reductions of sedentary time in office settings, the majority of participants have been women [7,10,11]. During the analysis of the interview data, no patterns for differences between men and women emerged, neither for behaviors, nor for their perceptions of the PA-promoting program. To address possible gender differences regarding responsiveness to interventions, more studies in male-dominated sectors would be of importance.

The study was conducted within one organization in Sweden with sit-stand tables as standard, and a structured program for health promoting activities already in place at baseline. Generalization of our results might be difficult, because every organization is unique, and local adjustments always need to be made. By providing a thorough description of the context, study design, analysis and by using quotations in the text, we have facilitated for the reader to interpret the transferability, and we believe the results are conceivable in organizations in the Scandinavian region, but also in a broader international perspective [34,35]. During a PE study, it is suggested as an ideal to act as a passive observer [20]. The main results from the AOD study were reported back to the organization during the study period, and even though the reports did not include any consultative elements, this might have influenced the employees and compromised the external validity of our results.

## 5. Conclusions

The PE contributed to helpfully explaining the effects of the PA-promoting program. Even though the intervention was successfully implemented, the intended improvements in PA behaviors were modest, probably due to the combination of the long-term availability of sit-stand desks and the strong organizational culture for posture variation and PA. Nevertheless, our study shows how a facilitating physical environment and strong organizational health culture can create sustainable positive behaviors for PA. Our results also show the complexity to fully fit the intervention to the context, and to meet the needs in the organization, despite a thorough participatory process during intervention development. The context for SB and PA behaviors in organizations can differ considerably, both in terms of culture, work tasks and physical environment, which indicate that intervention planning could, and probably should, vary between organizations. 

To tailor an intervention, both in research and in organizational practice, this should be based on a thorough understanding of the local conditions. PE studies are needed to fully understand outcome effects, and how to design effective interventions in different occupational contexts to decrease SB and increase PA.

## Figures and Tables

**Figure 1 ijerph-17-00023-f001:**
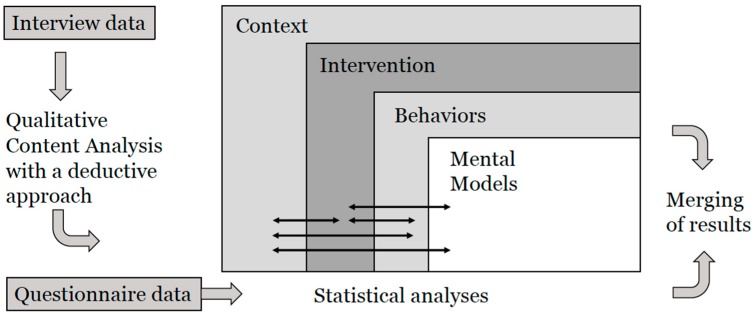
Process for data analysis and merging of qualitative and quantitative results.

**Figure 2 ijerph-17-00023-f002:**
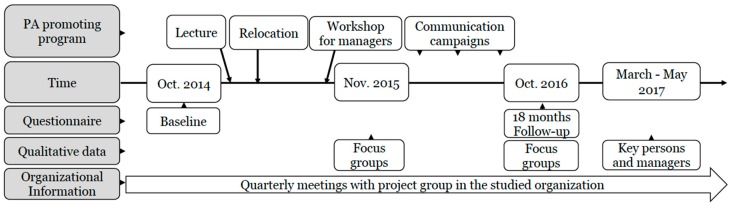
Flow chart for data collection and the physical activity program.

**Figure 3 ijerph-17-00023-f003:**
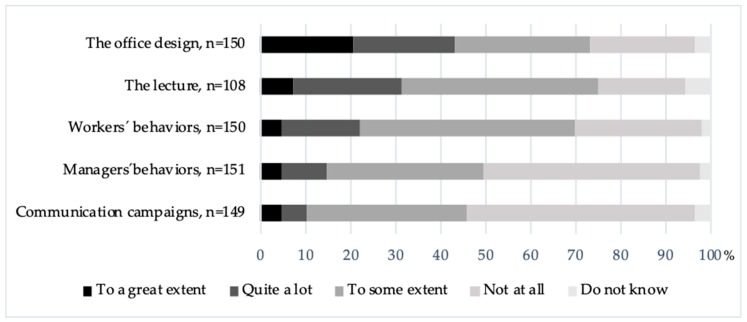
Motivational impact from program components to sit less and move more at work.

**Table 1 ijerph-17-00023-t001:** Baseline characteristics of employees (*n* = 152), 102 women and 50 men.

Demography and Work Characteristics	Women (%)	Men (%)	All (%)
Managerial position	25	28	26
Age groups			
18–39 years	28	26	28
40–49 years	24	30	26
>50 years	48	44	46
Employment degree			
100%	94	98	95
75–99%	6	2	5
Computer work per workday			
0–4 h	20	26	22
4–6 h	34	46	38
6–8 h	46	28	40
Meetings outside the office			
Never	23	20	22
1–2 times per month	28	36	31
3–4 times per month	22	18	20
1–2 times per week	17	16	16
3 times per week to daily	10	10	11
Standing while working individually			
Never	5	8	6
1–3 times per month	21	26	22
1–2 times per week	20	20	20
3–5 times per week	17	8	14
Daily	37	38	38
**Health and Lifestyle**			
Self-rated general health			
Very good or excellent	55	70	60
Good and fair	44	28	38
Bad	1	2	1
Physical exercise			
No exercise	2	6	3
Occasionally—not regular	24	22	23
Once a week	16	8	13
2–3 times a week	45	30	40
>3 times a week	14	34	21
Musculoskeletal neck-shoulder symptoms			
Never or seldom	46	60	51
Sometimes	29	20	26
Often	13	10	12
Always	12	10	11
Symptoms of stress			
Never or seldom	42	50	44
Sometimes	32	30	31
Often	22	14	19
Always	5	6	5

**Table 2 ijerph-17-00023-t002:** Subcategories and categories in relation to the elements in the process evaluation (PE)-model.

Subcategories	Categories	Element in PE-Model
Knowledge about PA and SB	Support for physical activity	Context
Worksite policy and routines		
Interior design		

Crowded office	Environmental and ergonomic challenges	
Workstation adjustments		
Role models	Clarity in the organization	Intervention
Responsibility at various levels		
Program ownership		
Focus in health promotion		
Strive for variation	Voluntary and involuntary physical activity	Behaviors
Walks to find a workplace		
Equipment and activities stimulate		

Productivity in focus	Mediators	
Physical discomfort		
Perception and interpretation	Balance of communication intensity	Mental models
of communication		
Conflicting needs		

Strive for physical activity	Openness for activity	
Necessities for progress		

**Table 3 ijerph-17-00023-t003:** Results of culture for being physically active (*n* = 152). Bold indicates statistically significant results.

**How frequent is it…**						
		**Never**	**Seldom**	**Sometimes**	**Often**	***p*-Value**
…with regular breaks from sitting during meetings? (%)	Baseline	23.2	33.8	36.4	6.6	**0.002**
	18 m	13.2	33.1	43.7	9.9	
…that you or a colleague suggest a walking meeting? (%)	Baseline	82.0	13.8	3.3	0.0	**0.001**
	18 m	69.5	22.5	7.9	0.0	
…that you or a colleague suggest a walk during breaks? (%)	Baseline	20.5	35.1	37.7	6.6	0.522
	18 m	28.5	27.8	35.8	7.9	
**To what degree…**						
		**Not at all**	**Somewhat**	**To some extent**	**To a big extent**	***p*-Value**
… is it socially accepted to stand or walk while working? ^1^ (%)	Baseline	5.4	12.8	11.4	70.5	**< 0.001**
	18 m	0.0	6.0	9.9	84.1	
… are your clothes a barrier for standing or walking at work? (%)	Baseline	88.1	4.0	7.3	0.7	0.062
	18 m	80.8	7.3	9.9	2.0	

^1^*n* = 149.

**Table 4 ijerph-17-00023-t004:** Results for usage and perceptions of physical activity possibilities at work at baseline and 18 months after relocation. Bold indicates statistically significant results.

**How often do you use the following possibilities to work standing or walking?**								
		**Never**	**1–3 times per month**	**1–2 times per week**	**3–5 times per week**	**Daily**	**NA ^1^ or missing**	***p*-Value**
Standing while working individually, %	Baseline	6	22.5	19.9	13.9	37.7	1	0.194
	18 m	10.0	20.7	22.0	12.0	35.3	2	
Standing at meetings, %	Baseline	84.1	11.0	3.4	0.7	0.7	7	**< 0.001**
	18 m	58.5	32.0	5.4	3.4	0.7	5	
Taking the stairs, %	Baseline	2.6	3.3	13.2	8.6	72.4	0	**< 0.001**
	18 m	1.3	0.7	2.0	2.7	93.3	3	
Participating in walking meetings, %	Baseline	87.4	10.5	1.4	0,0	0.7	9	0.689
	18 m	87.1	12.2	0.7	0.0	0.0	5	
Walk or cycle to meetings outside the office, %	Baseline	17.7	41.5	18.4	9.5	12.9	5	0.776
	18 m	16.7	44.4	15.3	13.2	10.4	7	
Treadmill station, %	18 m	87.9	9.4	2.0	0.7	0	3	
**How do you like the following possibilities to stand or walk at work?**								
		**Do not like at all**	**Do not like so much**	**Like it somewhat**	**Like a lot**	**No opinion ^2^**	**Missing**	***p*-Value**
Standing while working individually, %	Baseline	2.0	12.2	42.9	42.9	5	0	**0.003**
	18 m	6.2	19.3	37.2	37.2	5	2	
Standing at meetings, %	Baseline	39.1	23.4	23.4	14.1	88	0	0.056
	18 m	21.6	28.8	35.1	14.4	39	2	
Taking the stairs, %	Baseline	2.0	4.7	26.0	67.3	2	0	**< 0.001**
	18 m	1.3	0.0	8.7	90.0	1	1	
Participating in walking meetings, %	Baseline	22.8	21.1	29.8	26.3	95	0	0.122
	18 m	35.2	20.9	27.5	16.4	59	2	
Walk or cycle to meetings outside the office, %	Baseline	6.2	7.0	36.4	50.4	22	2	0.452
	18 m	6.6	2.9	33.8	56.6	14	2	
Treadmill station, %	18 m	44.7	15.8	9.2	2.6	39	3	

^1^ NA = Not applicable, ^2^ No opinion = I have tried too little to have an opinion.

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
