# Peer review of "Implementing a Physical Activity Promoting Program in a Flex-Office: A Process Evaluation with a Mixed Methods Design"

_ijerph, 2019, doi:10.3390/ijerph17010023_

Round 1
Reviewer 1 Report
Good study and a well-written paper. Few issues and suggestions:
The demographics of respondents are required to give more context to the findings reported. it would be helpful to show the influence of respondents' demography on their perception of the PA implementation process. When it comes to implementing changes in workplaces, research has shown that age, sex, and even culture play a huge part in the perception and effectiveness of the change. In the result section, it would help the reader if you start each discussion with the question asked (in the interviews or questionnaire). Having them on the tables/figures is not sufficient. The interview was semi-structured, please elaborate on the contents of the interview questions. this same goes with the questionnaire.Well done.
Author Response
Comments and Suggestions for Authors
Good study and a well-written paper. Few issues and suggestions:
The demographics of respondents are required to give more context to the findings reported. it would be helpful to show the influence of respondents' demography on their perception of the PA implementation process.
Response: Information about the departments with employees in the study are presented on page 3, row 100. Unfortunately, we do not have data to specify the work tasks within each department, but have added a brief information about within-group-variation and autonomy with respect to work tasks, on page 3, row 104.
As the qualitative data mostly is collected via focus groups it is not possible to extract and present the influence of participants demography on their perceptions of the PA-promoting program. An exception to this is the results related to the managers perspectives on the program, which is described in the result section on page 9, starting on row 312. Within the methodology of Qualitative content analysis there is a strive to present variations and patterns in the data. During the analysis of the interview data we found no patterns for differences between age-groups or gender regarding their perceptions of the implementation of the PA-promoting program. A clarification and description of this is added in the discussion section, page 16, row 607.
When it comes to implementing changes in workplaces, research has shown that age, sex, and even culture play a huge part in the perception and effectiveness of the change.
Response: Within the literature for sedentary behavior, and in intervention studies aiming to decrease sitting and increase physical activity, no major differences for age or gender have emerged. We have added the baseline descriptive variables for men and women separately in Table 1 (page 7), and described baseline differences on page 6, row 261. We have also added information about baseline values for age, page 6, row 263. As the groups are quite small (50 men and 102 women) and the changes in the quantitative data between baseline and follow-up are modest, we have not performed any subgroup analysis.
To our knowledge, in most intervention studies aiming to decrease sedentary behavior in office settings, there has been a majority of women participating. To address possible gender differences regarding responsiveness to interventions, more studies in male dominated sectors should preferably be performed. We have added this aspect in the discussion section, page 16, row 609. When addressing gender aspects, we also argue that it is of equal importance to address sameness between women and men as well as gender differences.
The culture is definitely an important factor for the effectiveness of the change. The detailed contextual information throughout the study is hopefully helpful to put the result in context and thereby facilitate the readers interpretation and transferability of results.
In the result section, it would help the reader if you start each discussion with the question asked (in the interviews or questionnaire). Having them on the tables/figures is not sufficient.
Response: In the result section the presentation of results is driven and presented with the categories that emerged from the qualitative analysis as a starting point. As the subcategories and categories emerge from the coded data (Table 2), the results are to some extent abstracted and do not answer a specific interview question. This makes it hard to start each discussion in the result section with a specific interview question.
In relation to the qualitative results, some of the questionnaire results are described in the text. In this way, the different types of data complement each other. A major challenge throughout the manuscript writing, has been to make the text readable and fluent. Even though it might become clearer adding the specific questions also in the result presentation (and not only in the tables) it would probably make the text less fluent to read. The descriptions of the questions used have now been somewhat clarified in the method section, page 5, section 2.3.2 Quantitative data.
The interview was semi-structured, please elaborate on the contents of the interview questions. this same goes with the questionnaire.
Response: The content of the semi-structured interview questions has been presented more in detail on page 4, row 165-167 and row 174.
The description of the questions in the questionnaire are clarified. Page 5, row 188-210.
Well done.
Response: Thanks!
Reviewer 2 Report
Comments to Authors:
Thank you for the opportunity to review this manuscript. The topic is relevant and studies in this topic are definitely needed. Therefore, the study is of interest. Based on the information provided by the authors, I have some reservations about the methodology of the manuscript. With the aim to improve the manuscript quality, I have raised some issues that need to be addressed.
General information: Different issues related to methodology should be checked. It would be interesting to improve the exposure aim. Furthermore, it could be clarified the size and sample characteristics. Moreover, I suggest to provide more information about the instrument used. Also, quantitative statistical analysis could be more specific.
Intro
-Line 35: the authors mention several “studies”, but only one study is cited. Are there more studies addressing this issue? Please cite them.
-Aim
The first study aim is “to evaluate the implementation process of a PA-promoting program using a theoretical model developed for PEs”. The key research question is not clear, please be more precise. Furthermore, the aim should be in line with the results and discussion. In discussion section the aim is described like “to conduct a PE of a PA-promoting program”.
Methods
-Line 93: Please specify the sample size of the overarching project and the sample size of actual study. I would like to know the method you followed to calculate sample size a priori.
-Line 102: Please indicate the exact number of employees. It should be clarified the differences between the workplaces before and after the flex-office.
-Line 106, 107: I would like to know whether this information has real interest in the method section: “Each floor had seven centrally placed waste-paper bins and two printer rooms”.
-Regarding to participants, the information is not clear, it could be interesting to show a flowchart with participants number, reasons to refuse, and the total number of measurements.
-Line 111: the overarching project has “around 400 participants”, what are the reasons to choose 228 for the questionnaire? Please indicate it at manuscript.
-Line 112, please clarify the reasons about the sample different between baseline and 18 months before data (flowchart).
-Line 116: Why is it relevant to inform about the manager's job? Please inform about the other participant jobs in overcharging project and in the present analysis (type and percentage).
-Line 116-121: information about focal group should be clarified. Which criteria did you follow to organize the people in groups?
-Line 130: table 1 should be inserted in the results section. This table could be more specific (demographic data, type of work).
-Line 132: the information of this paragraph is about the PA-promoting program of the overcharging project, so it could be more interesting to report this information in the 2.1 paragraph.
-Line 173: if questionnaires are evaluated 18 months before, why are there interviews conducted 6 months later? Please clarify it in the manuscript.
-Line 181: who are the key people? Please specify it in the manuscript.
-Line 182: why do the interviews take place 20 months later, and why are they individuals? Please clarify it in the manuscript.
-Line 194: it should be clarified the total amount number of measurements, in this line there are 3. This information differs from the participants section (line 113). Please clarify the questionnaire evaluations timing.
-Line 199, Please provide reliability data about the instrument.
-In the paragraph of quantitative data there is no information about the questionnaire.
-Reliability of instrument and PE should be reported.
-Line 224, it should be thoroughly described the fourth element.
-Line 228, It is necessary to report the instrument reliability.
-Line 245, non-parametric tests are used, but the normality test is not reported.
-Quantitative data statistical analysis is scarce (e. g, reliability and effect size).
Moreover, it is unclear the number of evaluations. In this case two have been analyzed (base line and 18 month), what about 6 months (before described in line 194)?
Results
-Line 261, is the subsidized gym supervised? Who is the institution that subsidizes it?
Discussion
-Line 428 (suggesting aforementioned): the aim of the study does not coincide with the method aim.
The aim of this section is described like “The aims of this study were to conduct a PE…” while in method section is defined like “to evaluate the implementation process”.
-Line 441: writing of this paragraph could be improved. “Which is in accordance with our objective measurements”: The results should be compared with the results from previous studies instead of other studies’ aims.
-Line 555: It would be of interest that this topic be addresed in “method” section and in “discussion” section should be highlighted like a potential limitation.
Author Response
Comments and Suggestions for Authors
Comments to Authors:
Thank you for the opportunity to review this manuscript. The topic is relevant and studies in this topic are definitely needed. Therefore, the study is of interest. Based on the information provided by the authors, I have some reservations about the methodology of the manuscript. With the aim to improve the manuscript quality, I have raised some issues that need to be addressed.
General information: Different issues related to methodology should be checked. It would be interesting to improve the exposure aim. Furthermore, it could be clarified the size and sample characteristics. Moreover, I suggest to provide more information about the instrument used. Also, quantitative statistical analysis could be more specific.
Intro
-Line 35: the authors mention several “studies”, but only one study is cited. Are there more studies addressing this issue? Please cite them.
Response: Thanks for addressing the need to cite more studies. More studies are added as references. Page 1, row 39.
-Aim
The first study aim is “to evaluate the implementation process of a PA-promoting program using a theoretical model developed for PEs”. The key research question is not clear, please be more precise. Furthermore, the aim should be in line with the results and discussion. In discussion section the aim is described like “to conduct a PE of a PA-promoting program”.
Response: The aim is clarified both in the method section on page 2, row 84, and in the beginning of the discussion section (page 13, row 462).
Methods
Line 93: Please specify the sample size of the overarching project and the sample size of actual study. I would like to know the method you followed to calculate sample size a priori.
Response: We have simplified the description of the sample size, by only describing the group that are studied in the current study. The number of employees in the flex office group at baseline are specified on (page 3, row 100).
As the study is a natural experiment, we performed no power calculations a priori for the questionnaire data. Power calculations were only performed for measurements and evaluations for objectively measured sedentary behavior and physical physical activity outcomes, presented in a previously published paper. Wahlström, V.; Bergman, F.; Öhberg, F.; Eskilsson, T.; Olsson, T.; Järvholm, L.S. Effects of a multicomponent physical activity promoting program on sedentary behavior, physical activity and body measures: a longitudinal study in different office types. Scand J Work Environ Health. 2019, 45, 493-504. doi:10.5271/sjweh.3808.
Line 102: Please indicate the exact number of employees. It should be clarified the differences between the workplaces before and after the flex-office.
Response: Thanks for your suggestion to clarify this aspect. All the information about the differences between the workplaces before and after the relocation is moved to the section for “Development and implementation of the physical activity promoting program”, on page 3, row 111. By presenting all information about the physical environment in the same section, I hope it is easier to grasp the differences in the office environments before and after the relocation.
-Line 106, 107: I would like to know whether this information has real interest in the method section: “Each floor had seven centrally placed waste-paper bins and two printer rooms”.
Response: This information is moved to the section for “Development and implementation of the physical activity promoting program”, as described in the response above.
-Regarding to participants, the information is not clear, it could be interesting to show a flowchart with participants number, reasons to refuse, and the total number of measurements.
Response: We have clarified the number of employees of interest for the study, and decided to only describe the group that has been evaluated in the current study (and not mention the control group in the AOD-study). Hopefully this makes it clearer. Page 3, row 100. We have also presented the number of participants, and reasons for drop-out or refuse. You find it in the result section on page 6, row 258.
-Line 111: the overarching project has “around 400 participants”, what are the reasons to choose 228 for the questionnaire? Please indicate it at manuscript.
Response: We have clarified the number of employees of interest for the study, and decided to only describe the group that has been evaluated in the current study (and not mention the control group in the AOD-study). Hopefully this makes it clearer. Page 3, row 100.
-Line 112, please clarify the reasons about the sample different between baseline and 18 months before data (flowchart).
Response: In the result section (Page 6, row 258) we have added information about reasons for drop-out between baseline and 18 months follow-up. As there is already several figures and tables in the manuscript, we prefer not to add a flow-chart, but if you find it important, we could of course also provide a flowchart of the sample, with reasons for drop-out and non-respondents.
-Line 116: Why is it relevant to inform about the manager's job? Please inform about the other participant jobs in overcharging project and in the present analysis (type and percentage).
Response: As managers are crucial for the implementation of an organizational intervention, we find it important to describe that interviews among managers were performed, page 6, line 265.
Information about the departments with employees in the study are presented on page 3, row 100. Unfortunately, we do not have data to specify the work tasks within each department, but have added a brief information about within-group-variation and autonomy in relation to work, page 3, row 102.
-Line 116-121: information about focus group should be clarified. Which criteria did you follow to organize the people in groups?
Response: An invitation was sent to all employees, and recruitment to interviews was based on a convenient sampling from the response to these invitations. This is clarified on page 4, row 159. At 6 months follow up, managers and other employees participated in separate focus groups, which is clarified on page 6, row 266. There was no other specific criteria for the organization of participants in the focus groups.
-Line 130: table 1 should be inserted in the results section. This table could be more specific (demographic data, type of work).
Response: We have moved Table 1 to the result section, and added the variables (musculoskeletal neck-shoulder symptoms, stress, and self-reported standing while working individually). All background variables are presented for women, men and all respondents. We also tested for differences for age and sex for the baseline variables, and have added brief information about this in the result section, on page 6, row 261-264. Unfortunately, we do not have data to specify the work tasks within each department, but have added a brief information about within-group-variation and autonomy with respect to work tasks, on page 3, row 104.
-Line 132: the information of this paragraph is about the PA-promoting program of the overcharging project, so it could be more interesting to report this information in the 2.1 paragraph.
Response: Thanks for your advice. The section describing the PA-promoting program is moved and is reported as 2.2 following the information of the overarching project. Page 3, row 105-144.
-Line 173: if questionnaires are evaluated 18 months before, why are there interviews conducted 6 months later? Please clarify it in the manuscript.
Response: Questionnaires are evaluated for 6 months before and 18 months after the relocation. Both focus group periods were conducted after the relocation, at 6 and 18 months after the relocation. The timepoints for the focus groups are described in the text on page 4, row 159. and is shown in Figure 2.
-Line 181: who are the key people? Please specify it in the manuscript.
Response: The description of the key persons is specified. Page 6, row 270.
-Line 182: why do the interviews take place 20 months later, and why are they individuals? Please clarify it in the manuscript.
Response: Thanks for highlighting the lack of this information. The reason for conducting individual interviews at a later stage are clarified on page 6, row 268.
-Line 194: it should be clarified the total amount number of measurements, in this line there are 3. This information differs from the participants section (row 113). Please clarify the questionnaire evaluations timing.
Response: Thanks for your suggestion. As we have not analyzed and used the questionnaire data from 6 months after the relocation, we have removed the information about this data collection period, not to create confusion. Thereby we only describe the questionnaire data collection from 6 months before and 18 months after the relocation.
-Line 199, Please provide reliability data about the instrument.
Response: Reliability data of the question for self-reported health is reported, Page 5, row 192.
-In the paragraph of quantitative data there is no information about the questionnaire.
Response: Information about the lack of valid instruments for the questionnaires are added in the manuscript, following your suggestion, page 5, row 197.
-Reliability of instrument and PE should be reported.
Response: Using a theoretical model when performing a process evaluation is recommended as it provides a systematic and thorough basis for the data collection and evaluation. The model and the related framework developed by Nielsen et al, worked as a guidance during data collection, analyses and reporting of results. The framework includes aspects that are important to consider during the different steps of the process evaluation. Thus, the theoretical model is not a questionnaire instrument, and cannot be evaluated for reliability.
-Line 224, it should be thoroughly described the fourth element.
Response: The fourth element is now more thoroughly described on page 6, row 225-229.
-Line 228, It is necessary to report the instrument reliability.
Response: We used the method Qualitative Content Analysis to analyze the interview data in a structured way. It is not possible to report reliability of this method, instead we have used the overarching qualitative term trustworthiness including more specific terms (credibility, conformability, dependability) to present to the reader how we have used this qualitative method. For qualitative content analysis trustworthiness could be described and strengthened in different ways. In this case, credibility was strengthened by including many employees, men and women, key persons as well as managers in the interviews. To improve the trustworthiness of the results, we have presented information about how and when we collected the qualitative data, and who participated in the interviews. By analyzing interview data from both 6 and 18 months after the relocation, the dependability of results over time is strengthened. To further strengthen the analyzed data, we had internal discussions within the author group, as well as with other researchers involved in the AOD-study research group. These discussions are also a way to strengthen the conformability of the results. To facilitate the readers interpretation of the transferability of the results, we have thoroughly described the contextual details in the organization.
-Line 245, non-parametric tests are used, but the normality test is not reported.
Response: Due to the categorical data we have used Wilcoxon matched paired tests. As we have ordinal data, without obvious numerical values, we don´t find it motivated to describe the distribution of the differences, according to a continuous distribution.
-Quantitative data statistical analysis is scarce (e. g, reliability and effect size).
Response: In the paper the qualitative results are driving the presentation, why we have chosen to present the quantitative results in a descriptive manner, by reporting the distributions of the scores from the questionnaires at each time point. This descriptive presentation makes it possible for the reader to interpret how the changes in scores differed between baseline and 18 months. As we used 4- or 5-graded scales it is difficult to assess the effects size, and whether the changes are clinically relevant.
Moreover, it is unclear the number of evaluations. In this case two have been analyzed (base line and 18 month), what about 6 months (before described in line 194)?
Response: Thanks for your clear eyes. We have clarified this by removing information about the questionnaires from 6 months after baseline.
Results
-Line 261, is the subsidized gym supervised? Who is the institution that subsidizes it?
Response: The studied organization subsidized the gym fees for external gyms. This is common in Sweden and there are usually no rules for the external fitness centers to be supervised, even though most of them are. We have tried to clarify this in the manuscript, page 8, row 288.
Discussion
-Line 428 (suggesting aforementioned): the aim of the study does not coincide with the method aim.
The aim of this section is described like “The aims of this study were to conduct a PE…” while in method section is defined like “to evaluate the implementation process”.
Response: Thanks for highlighting this limitation. The description of the first aim of the study is adjusted in the discussion section and harmonized with the method aim.
-Line 441: writing of this paragraph could be improved. “Which is in accordance with our objective measurements”: The results should be compared with the results from previous studies instead of other studies’ aims.
Response: The paragraph is updated. As we know that self-reported sitting is less valid than objective measurements, we argue that it is of interest to compare and discuss the current self-reported findings with our previously performed and reported results using objective measurements from the same project. Following this, our results are discussed in comparison to other studies.
-Line 555: It would be of interest that this topic be addressed in “method” section and in “discussion” section should be highlighted like a potential limitation.
Response: This information is added in the method section (Page 5, row 196), and is instead only highlighted as a potential limitation in the discussion (Page 16, row 594).
Round 2
Reviewer 2 Report
Thank you for considering my suggestions. The manuscript is more appropriate now.
Nevertheless, I would like to add that the aim of study must be more precise. With the new update (row 82), the aim is not clear. I think it could be simplified.
Author Response
Comments and Suggestions for Authors.
Thank you for considering my suggestions. The manuscript is more appropriate now.
Nevertheless, I would like to add that the aim of study must be more precise. With the new update (row 82), the aim is not clear. I think it could be simplified.
Response: Thanks, for your feedback. We have simplified the description on the aim, to make it more clear.
Small updates and improvements in terms of language is also done in the manuscript.